# Enhancement of Radiation Therapy through Blockade of the Immune Checkpoint, V-domain Ig Suppressor of T Cell Activation (VISTA), in Melanoma and Adenocarcinoma Murine Models

**DOI:** 10.3390/ijms241813742

**Published:** 2023-09-06

**Authors:** Kayla E. A. Duval, Armin D. Tavakkoli, Alireza Kheirollah, Haille E. Soderholm, Eugene Demidenko, Janet L. Lines, Walburga Croteau, Samuel C. Zhang, Robert J. Wagner, Ethan Aulwes, Randolph J. Noelle, P. Jack Hoopes

**Affiliations:** 1Department of Surgery, Geisel School of Medicine, Hanover, NH 03755, USA; kaeduval@gmail.com (K.E.A.D.); armin.d.tavakkoli.med@dartmouth.edu (A.D.T.); alireza.kheirollah@dartmouth.edu (A.K.); haille.e.soderholm.gr@dartmouth.edu (H.E.S.); sam.c.zhang.med@dartmouth.edu (S.C.Z.); eaulwes@outlook.com (E.A.); 2Department of Biomedical Data Science, Geisel School of Medicine, Hanover, NH 03755, USA; eugene.demidenko@dartmouth.edu; 3Department of Microbiology and Immunology, Geisel School of Medicine, Hanover, NH 03755, USA; janet.lines@dartmouth.edu (J.L.L.); randolph.j.noelle@dartmouth.edu (R.J.N.)

**Keywords:** checkpoint inhibitor, VISTA, radiation therapy

## Abstract

Radiation therapy (RT) has recently demonstrated promise at stimulating an enhanced immune response. The recent success of immunotherapies, such as checkpoint inhibitors, CART cells, and other immune modulators, affords new opportunities for combination with radiation. The aim of this study is to evaluate whether and to what extent blockade of VISTA, an immune checkpoint, can potentiate the tumor control ability of radiation therapy. Our study is novel in that it is the first comparison of two VISTA-blocking methods (antibody inhibition and genetic knockout) in combination with RT. VISTA was blocked either through genetic knockout (KO) or an inhibitory antibody and combined with RT in two syngeneic murine flank tumor models (B16 and MC38). Selected mRNA, immune cell infiltration, and tumor growth delay were used to assess the biological effects. When combined with a single 15Gy radiation dose, VISTA blockade via genetic knockout in the B16 model and via anti-VISTA antibodies in the MC38 model significantly improved survival compared to RT alone by an average of 5.5 days and 6.3 days, respectively (*p* < 0.05). The gene expression data suggest that the mechanism behind the enhanced tumor control is primarily a result of increased apoptosis and immune-mediated cytotoxicity. VISTA blockade significantly enhances the anti-tumor effect of a single dose of 15Gy radiation through increased expression and stimulation of cell-mediated apoptosis pathways. These results suggest that VISTA is a biologically relevant immune promoter that has the potential to enhance the efficacy of a large single radiation dose in a synergic manner.

## 1. Introduction

Radiation therapy (RT) is a primary treatment for various cancers, often in combination with surgery or chemotherapy. Classical RT fractionation can stimulate an immune response; however, studies suggest that hypofractionated RT can stimulate an even stronger immune response [1,2]. Unfortunately, hypofractionation also triggers various immune suppressive signals that can limit the anti-cancer immune response, which provides a potential role for the use of immune modulators [1,2,3,4]. Combining radiation with immunotherapy would allow for enhancement of the radiation effect without increasing the radiation dose.

One such immunotherapy option is an immune checkpoint inhibitor. Immune checkpoint therapies have demonstrated dramatic but varied success in the past decade, with programmed cell death protein 1 (PD-1)/associated ligand (PD-L1) and cytotoxic T-lymphocyte-associated protein 4 (CTLA-4) being the most frequently used and researched [5,6]. PD-1 and CTLA-4 blockade prevent T-cell suppression, thus potentiating the anti-tumor response. These therapies have demonstrated promise alone and in combination with radiation, but their effects are highly variable [6,7]. There is recent evidence that the lack of PD-1/CTLA-4 blockade efficacy can occur through upregulation of alternative immune checkpoints [8]. Several other immune checkpoints have been recently identified, including VISTA (V-domain Ig Suppressor of T cell Activation) [9]. VISTA is expressed on hematopoietic and myeloid-derived cells, tumor cells, as well as various T cell populations [10,11,12,13]. The presence of VISTA on both antigen-presenting cells and regulatory T cells appears to be most important for limiting cytotoxic T cell activity and function [14]. VISTA has both receptor and ligand functions that act by suppressing and negatively regulating the activation and function of T cells, while also promoting immune suppression through expression on myeloid-derived suppressor cells and regulatory T cells [15,16]. Anti-VISTA has shown immune stimulatory and tumor control promise when used alone, and in combination with anti-PD-1 and anti-CTLA4 [17,18].

The aim of this study is to evaluate whether and to what extent VISTA blockade can potentiate the tumor control ability of radiation therapy at the same dose, and to evaluate the potential underlying mechanisms of this enhancement. In this study, we hypothesize that combining VISTA inhibition with RT will result in an enhanced T-cell-mediated apoptosis response and overall improved treatment efficacy in B16F10 melanoma and MC38 adenocarcinoma murine flank tumor models. In separate studies, we utilize VISTA knockout (KO) mice and anti-VISTA antibodies to achieve VISTA blockade and combine this with a single dose of 15Gy radiation. The amount of 15Gy is a frequently employed dose in radiation research and has been shown to be relevant for the assessment of RT and immune modulation studies. Although several promising combined radiation and immune checkpoint inhibition studies have been published, our study is novel in that it is the first comparison of two VISTA-blocking methods (antibody inhibition and genetic knockout) in combination with RT. Furthermore, our study is unique in its use of NanoString gene expression analyses to characterize the underlying mechanisms of RT enhancement through VISTA blockade.

## 2. Results

### 2.1. VISTA-KO + RT Leads to Improved Tumor Control in the Murine B16F10 Flank Tumor Model

Using the previously described tumor growth endpoint (days to tripling volume), WT and VISTA-KO animals remained on study an average of 3.3 days (SEM = 0.3) and 3.43 days (SEM = 0.2), respectively. The WT/15Gy and VISTA-KO/15Gy animals remained on study an average of 12 days (SEM = 1.39) and 17.5 days (SEM = 1.95), respectively. Using a two-way ANOVA, we determined that there was no significant difference between the WT and VISTA-KO control cohorts. However, both radiation cohorts were significantly different from controls (*p* < 0.037). Most importantly, the VISTA-KO/15GY radiation cohort significantly outperformed the WT/15GY radiation cohort (*p* < 0.02). These data are summarized in Figure 1.

To confirm this effect, we repeated the VISTA-KO study using a murine colon adenocarcinoma MC38 cell line. The Kaplan–Meier treatment efficacy data are shown in Figure 2. Similar to the B16F10 model, there was no significant difference between the WT and VISTA-KO control arms with an average survival of 6.5 days (SEM = 0.41) and 8.13 days (SEM = 1.06), respectively. Comparatively, the WT/15Gy group remained on study an average of 16 days (SEM = 1.93) and the VISTA-KO/15Gy group remained an average of 19.3 days (SEM = 1.8). Although these cohorts were not statistically different, a trend toward significance was present. Additionally, tumor growth rates for each treatment were determined to gain a better understanding of the differences and effects between cohorts. As expected, the growth rate of all irradiated tumors was slower than that of the unirradiated control; however, there were minimal differences between the WT and KO radiation cohorts. In this model, it appeared that radiation was equally effective at tumor control with or without VISTA KO blockade.

### 2.2. Anti-VISTA Antibody Blockade and RT Improve Tumor Control in a B16 Melanoma Model

Following assessment in the VISTA-KO model, a similar study was designed using an anti-VISTA antibody blockade and +/− RT model. PBS, isotype antibody control, and anti-VISTA blockade were used with or without radiation in the B16 melanoma model. Kaplan–Meier data are shown in Figure 3. The unirradiated PBS, isotype, and VISTA blockade averaged 4 (SEM = 0.47), 4 (SEM = 0.33), and 5 (SEM = 0.37) days on study, respectively. The radiated groups survived significantly longer, with isotype/15Gy and Anti-VISTA/15Gy remaining on study an average of 11.5 days (SEM = 1.39) and 17.8 SEM = 1.05) days, respectively. Using a two-way ANOVA of survival, PBS and unirradiated isotypes and anti-VISTA cohorts were not statistically different from each other. Both 15Gy cohorts (isotype and anti-VISTA) survived significantly longer than unirradiated groups (*p* < 0.05). The isotype/15Gy and VISTA blockade/15Gy groups did not differ in survival from each other (*p* < 0.0003).

In summary, when combined with a single 15Gy dose of radiation, VISTA blockade via genetic knockout in the B16 model and via anti-VISTA antibodies in the MC38 model significantly improved survival by an average of 5.5 days and 6.3 days compared to RT alone, respectively (*p* < 0.05). VISTA blockade via genetic knockout in the MC38 model improved survival by an average of 3.3 days compared to RT alone, but this difference was not statistically significant. 

### 2.3. Anti-VISTA Antibody Leads to Anti-Tumor mRNA Expression Changes

For the anti-VISTA antibody study, we used NanoString technologies quantification analytics to assess changes in RNA expression in the B16 melanoma tumor. Unless otherwise specified, the comparative anti-VISTA genetic changes noted below are compared to an isotype antibody control.

Following anti-VISTA antibody treatment, two genes in the immune cell adhesion/migration and cell signaling/metastasis pathways changed significantly (defined as greater than a twofold change and *p* < 0.05): genes Clec4e (C-type lectin domain family 4 member E, cell adhesion molecule) and Ptger4 (Prostaglandin EP4 receptor, T-cell signaling).

When the VISTA blockade was combined with 15Gy radiation, genes associated with the following pathways were affected: antigen processing and presentation (both MHC I and MHC II), T cell receptor signaling (including CD28, CD3, and CD247), and leukocyte trans-endothelial migration (including ITGAL, ITGB2, ICAM1, VCAM1, and PECAM1). Although many immune and cytotoxic pathways were expressed higher by both radiation groups (anti-VISTA/15Gy and isotype control/15G), several notable genes were enhanced with the addition of anti-VISTA antibody (Ccl2, Cxcl12, Siglec1, and Tnfsf13). Expression level details are presented in Table 1.

### 2.4. Immune Infiltration Based on mRNA Expression

The RNA data also demonstrate that VISTA blockade with radiation leads to an increased cytotoxic immune cell presence. The NanoString program has a cell type profiling analysis and allows for a qualitative comparison of relative cell abundance in the B16 melanoma tumor between different treatment types (Figure 4). The data suggest that a combination of 15Gy/VISTA blockade leads to a slightly higher tumor level of cytotoxic immune cells such as CD8 T cells and Natural Killer cells, as compared to the 15Gy/isotype control. Additionally, 15Gy/VISTA blockade demonstrated a higher ratio of functional CD8+ cells to exhausted CD8+ cells.

## 3. Discussion

Radiation therapy (RT) is an effective cancer therapy. However, due to a variety of patient and normal tissue complications and its stimulation of both pro- and anti-tumor signaling, it is often not curative. Immune checkpoints are among the pro-tumor signals associated with RT. For this reason, we evaluated whether combining radiation with the anti-VISTA checkpoint inhibitor could enhance the therapeutic effect of RT. To assess this hypothesis, flank tumors were grown in mice using two well-studied syngenetic murine tumor lines (B16F10 and MC38), and VISTA blockade was achieved using either the genetically modified VISTA-KO mice or anti-VISTA antibody in WT mice.

The VISTA-KO and anti-VISTA blockade resulted in significantly longer survival in the B16 and MC38 models by an average of 5.5 and 6.3 days compared to RT alone, respectively (*p* < 0.05). The studies also suggest that the blockade of VISTA enhances the anti-tumor effect of radiation in a synergistic manner. If the effect were additive rather than synergistic, there would be a marked difference in the cohorts without radiation. The single dose of radiation combined with a systemic administration of VISTA antibody blockade demonstrated a significant enhancement in tumor control, indicating that VISTA blockade can be used safely and effectively. 

Although the mechanism underlying the enhancement of RT through VISTA blockade is being investigated, our mRNA and immune infiltration data strongly suggest that the enhancement is associated with cell-mediated apoptosis. For the VISTA antibody blockade alone, the most impressive mRNA changes were Clec4e and Ptger4. The function of Clec4e has been debated, with respect to its pro-inflammatory/immune and immune regulatory functions [19,20]. Its role could either support pro-immune activation through the innate pathway, or it could indicate new pathways of immune regulation following a VISTA block. This evasion characteristic has been seen with other immune checkpoint inhibitors, as reported in previous studies [9,21]. On the other hand, Ptger4, a subtype of the prostaglandin E receptor, has been shown to be involved in the metastasis pathway. Studies utilizing antagonists and KO models have demonstrated a decrease in metastatic potential and decreased tumor growth [22,23]. This suggests that a decrease in Ptger4 following VISTA blockade could result in a longer-term anti-tumor effect. As expected, the combination of VISTA blockade with radiation stimulated many mRNA expression changes in various immune and cytotoxic pathways. There were several notable trends and differences between the two radiation cohorts that could indicate an enhanced effect at the genetic or immune cell infiltration level. CD247 and CD3 had higher expression changes with the addition of VISTA antibody blockade to RT than did the isotype control, indicating increased T cell stimulation and activation. Additionally, cytotoxic genes, such as Fas Ligand and Granzyme B, had higher expression changes, further implicating immune-cell-mediated cytotoxicity.

Together, these mRNA changes provide evidence that VISTA blockade, especially with 15Gy radiation, is capable of initiating a significant anti-tumor genetic change. It is worth noting that anti-VISTA antibodies, as well as other checkpoint inhibitors, act at the protein level to prevent the immune checkpoint/regulatory function associated with VISTA. Any changes would seem to indicate that the treatment had enough effect on a cell structure or process that translated to an alteration in RNA expression levels. 

The assessment of tumor immune cell infiltration through the mRNA expression of characteristic genes demonstrated several potentially important anti-tumor signals. One important example is that VISTA blockade alone or combined with radiation led to increased tumor cytotoxic T cells and NK cells. In addition, there was an enhanced ratio of CD8 T cells to exhausted T cells, indicating an enhanced cytotoxic immune presence within the tumor. This finding further supports an increase in immune-mediated apoptosis, and better overall tumor control.

Several limitations of our study should be discussed. In our study, radiation therapy alone showed significant tumor control for all study models. However, as expected, the response to radiation was tumor-model-specific, with MC38 tumors being slightly more resistant to radiation. The VISTA-KO and anti-VISTA blockade resulted in significantly longer survival in the B16 and MC38 models. Although VISTA blockade via genetic knockout improved survival in the MC38 model by an average of 3.3 days, this difference was not statistically significant. This is likely attributed to the lack of sufficient statistical power in our study and overall therapy resistance of the MC38 model. However, as noted, the ability of anti-VISTA antibodies to significantly improve survival even in this more resistant model is reassuring, since VISTA blockade via antibodies is the more translationally relevant approach compared to genetic knockout. 

An interesting feature of the VISTA/radiation treatment is the heterogeneity of the individual response. In our study, we saw evidence that some animals who received VISTA blockade and RT showed a marked tumor control response, whereas others showed a lesser or minimal response. This distinction between “responders” and “non-responders” is characteristic of existing immune therapies and is observed with VISTA blockade as well. We therefore hypothesize that the treatment combination may be especially beneficial to responders compared to non-responders. As such, the identification of responders at an early stage of cancer may dramatically improve the outcome.

In conclusion, these experiments have demonstrated for the first time that VISTA blockade significantly enhances the therapeutic effect of radiation in two murine flank tumor models (B16 and MC38). Furthermore, NanoString genetic expression analyses show that the enhancement of RT is through increased immune-mediated pathways, such as T-cell-induced apoptosis. Further studies are needed to determine if the significantly enhanced tumor control demonstrated by the combination of RT and VISTA blockade in our tumor models can translate to a clinically meaningful difference in survival in larger animal models and human patients.

## 4. Materials and Methods

### 4.1. Mouse Flank Tumor Models

In this study, we grew flank tumors in mice using two different cell lines, B16F10 murine melanoma and MC38 murine colon adenocarcinoma, obtained from American Type Culture Collection (ATCC) in Manassas, VA, USA. The B16F10 cells were cultured in RPMI media supplemented with 10% fetal bovine serum and 1% antibiotics until inoculation. The MC38 cells were cultured in DMEM media with 10% fetal bovine serum, and 1% antibiotics until inoculation. For VISTA knockout (KO) experiments, tumors were implanted in male C57BL/6 WT and C57BL/6 VISTA-KO mice. For the anti-VISTA antibody experiments, tumors were implanted in female WT C57BL/6 mice. Knockout mice and anti-VISTA antibody were acquired from Dr. Noelle’s laboratory at the Dartmouth Geisel School of Medicine and Immunext Corp., and wildtype mice were acquired from Charles River Corp. (Wilmington, MA, USA).

Tumors were implanted by injecting B16F10 cells or MC38 cells (2 × 10^6^) intradermally into the right flank. Radiation and immunotherapy treatments were initiated when a tumor reached 110 (+/−20) mm^3^. Mice were randomly placed in treatment groups, and tumors were measured daily by investigators blinded to treatment condition. Tumor length (l), width (w), and depth (d) were measured using calipers and tumor volume calculation followed the common ellipsoid approach where
tumor volume (mm^3^) = 1/6 π × l (mm) × w (mm) × d (mm).

Experimental VISTA-KO groups consisted of both B16 and MC38 tumor models and included WT control, KO control, WT/15Gy, and VISTA-KO/15Gy. Experimental antibody groups consisted of only the B16 tumor model and included PBS control, isotype control, anti-VISTA antibody, isotype/15Gy, and anti-VISTA/15Gy.

### 4.2. RT Delivery

Following a computerized treatment and delivery plan, a Varian 2100 CD Linear Accelerator was used to deliver a uniform 6 MeV tumor dose of 15Gy (SSD = 100 cm). The treatment encompassed the entire tumor and a 2 mm peritumor region. Mice were anesthetized with isoflurane for treatment and positioned using conventional radiation oncology laser beam location techniques.

### 4.3. Isotype Control Antibody and Anti-VISTA Antibody

The isotype control antibody (PIP monoclonal antibody, inVivoMAB Armenian hamster IgG isotype control, anti-glutathione S-transferase) was acquired from Bxcell (BE0260). This control antibody was selected due to the absence of glutathione S-transferase expression in mammals. Both isotype control antibody and anti-VISTA antibody (13F3, Immunext Corp., Lebanon, NH, USA) treatments were given 3 times on consecutive days at 0.015 mg/g (diluted in PBS, injected in 0.3 mL IP). Antibody injection began immediately after a 15Gy radiation dose was delivered.

### 4.4. In Vivo Endpoints

Two cohorts of mice were used for each treatment group, with the first being sacrificed at 6 days post treatment for mRNA analysis (*n* = 5). The second group was sacrificed when the tumor reached 3 times its treatment volume for assessment of tumor growth delay (*n* = 8–10). Kaplan–Meier curves were constructed using “survival” values calculated using time to tumor volume tripling.

### 4.5. RNA Collection and Analysis

RNA isolation was performed using the Qiagen RNeasy Mini Kit. Normalized RNA samples were prepared, followed by mRNA expression quantification using the NanoString PanCancer IO 360 (murine) panel. This panel includes genes involved in the tumor biology, the tumor microenvironment, and the immune response. Expression values were then analyzed using nSolver Analysis software (v. 4.0) and Advanced Analysis Software (v. 2.0).

### 4.6. Statistical Analysis

Genetic/RNA (NanoString Inc, Seattle, WA, USA) assessment using nSolver Advanced Analysis utilizes XQuartz and R statistical software. ANOVA (in MATLAB) was used to analyze the Kaplan–Meier survival curve and tumor growth. Tumor growth data are displayed on a log scale and were fit with a mixed model, with *p*-value reported [24,25]. This advanced methodology accounts for the heterogeneity of individual response to treatment with the rate of tumor growth as the endpoint of the treatment effect. Unlike the traditional mean +/− SEM display, we show individual growth curves that allow assessment of the individual treatment response. Additionally, synergy was tested to determine if the survival/tumor growth for two treatments acts antagonistically or synergistically [26].

## Figures and Tables

**Figure 1 ijms-24-13742-f001:**
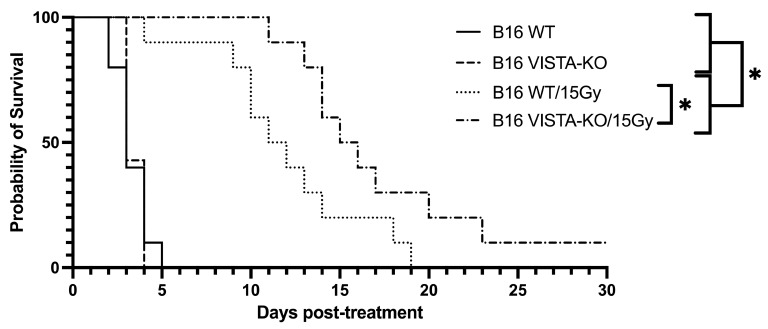
Kaplan–Meier curve demonstrating tumor control of flank B16 in the VISTA KO model. Kaplan–Meier survival curves are shown for wildtype control (WT Control), VISTA knockout control (VISA KO Control), wildtype + 15Gy radiation (WT/15Gy), and VISTA knockout + 15Gy radiation (KO/15Gy) groups. WT Control and VISTA KO Control groups do not differ in survival. Irradiated groups (WT/15Gy and VISTA-KO/15Gy) survived significantly longer than either non-irradiated control (*p ≤* 0.037), and VISTA-KO/15Gy improved survival significantly compared to WT/15Gy (*p <* 0.05). ** = p* < 0.05.

**Figure 2 ijms-24-13742-f002:**
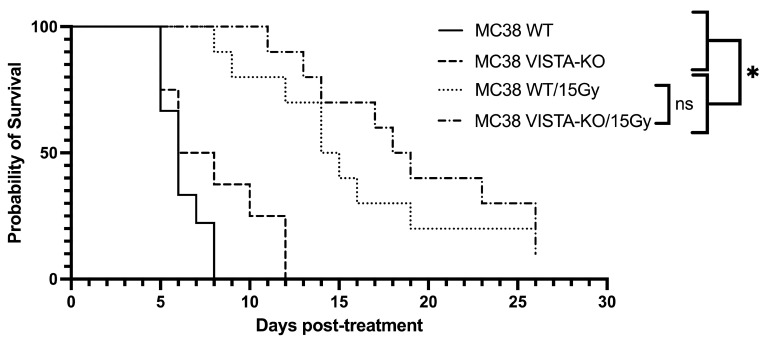
Kaplan–Meier curve demonstrating tumor control of flank MC38 in the VISTA KO model. Kaplan–Meier survival curves are shown for wildtype control (WT Control), VISTA knockout control (VISA KO Control), wildtype + 15Gy radiation (WT/15Gy), and VISTA knockout + 15Gy radiation (KO/15Gy) groups. WT Control and VISTA KO Control groups do not differ in survival. WT/15Gy and VISTA-KO/15Gy are both significantly different from either control (*p* < 0.05), but not from each other. * = *p* < 0.05, ns = not significant (*p* > 0.05).

**Figure 3 ijms-24-13742-f003:**
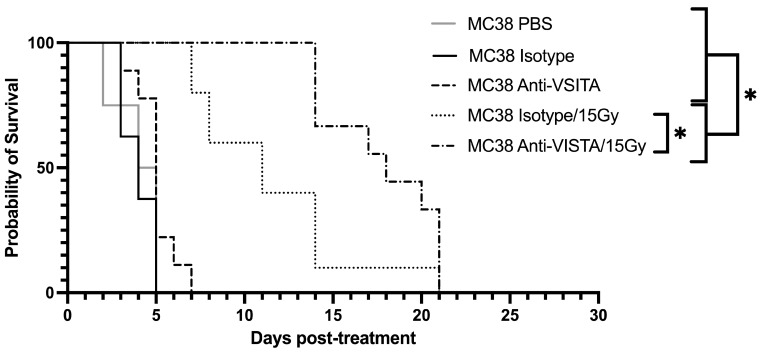
Kaplan–Meier curve demonstrating tumor control of flank MC38 in the anti-VISTA model. Kaplan–Meier curve for mice for either isotype control or anti-VISTA antibody with or without 15Gy radiation is shown, in addition to a PBS control group. Irradiated isotype and anti-VISTA groups differ significantly from non-irradiated groups. Moreover, the isotype/15Gy and anti-VISTA/15Gy differ significantly with *p* < 0.01. ** = p* < 0.05.

**Figure 4 ijms-24-13742-f004:**
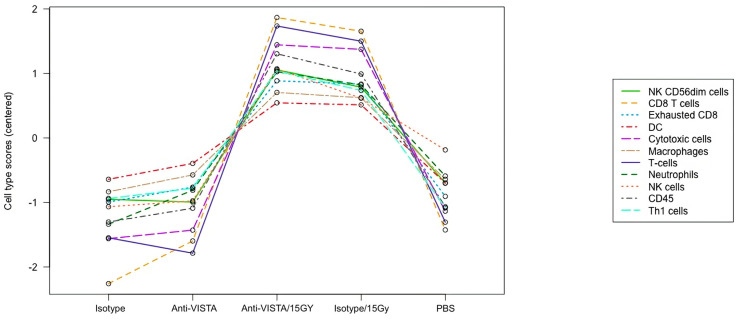
Immune cell abundance (mRNA) in B16 melanoma tumor. The RNA data (mRNA) demonstrate that VISTA blockade with radiation leads to increased cytotoxic immune cell presence in B16 melanoma tumor. Raw cell scores for an array of immune cell types for the different treatment cohorts. Activated T cells and NK cells (CD8 and CD56dim) were slightly higher in the anti-VISTA/15Gy than the isotype/15Gy cohort.

**Table 1 ijms-24-13742-t001:** Anti-VISTA antibody affects the RT-induced anti-tumor gene expression.

Gene	Anti-VISTA	Isotype Control + 15Gy	Anti-VISTA + 15Gy	Pathway/Function
Fasl	---	20.5 *	26.5 *	Fas ligand—apoptosis pathway
Clec4e	5.0 *	3 *	2.9	C-type lectin domain 4, member E
Ptger4	0.5 *	3.5 *	3.9 *	Supports cancer growth—knockouts prevent tumor growth
CD28	1.6	5.9 *	5.7 *	T cell receptor signaling pathway
CD3d/e/g	0.8/0.5/0.7	9.4 */8.9 */8.6 *	10.9 */10.6 */10.0 *	CD3 antigens, T cell receptor signaling pathway
CD247	1.6	14.9 *	22.1 *	T-cell receptor signaling (CD3-zeta)
ITGAL	1.2	6.6 *	8.2 *	Integrin alpha 1
ITGB2	0.9	3.1 *	3.4 *	Integrin beta 2
ICAM1	1.0	2.2 *	2.8 *	Intercellular adhesion molecule 1
VCAM1	1.4	2.0 *	2.6 *	Vascular cell adhesion molecule 1
PECAM1	1.1	1.8 *	1.8 *	Platelet/endothelial cell adhesion molecule 1
Cccl2	1.6	2.0 *	2.1 *	Chemokine ligand 2
Cxcl12	1.1	1.5	2.1 *	Chemokine (CXC motif) ligand 12
Siglec1	0.5 *	1.9 *	2.4 *	sialic acid binding Ig-like lectin 1, sialoadhesin, macrophage cell adhesion
Tnfsf13	1.8 *	1.5	2.3 *	Tumor necrosis factor ligand superfamily, member 13
Gzma/Gzmb	0.5/0.6	3.6 */5.5 *	3.1 */6.3 *	Granzyme A/Granzyme B—T-cell-mediated apoptosis

Differential expression data (as average linear fold change) for each treatment cohort, as compared to isotype control for notable genes as mentioned in the text. While anti-VISTA did not change many genes, it did affect Clec4e, Tnsfsf13, Ptger4, and Siglec1 significantly. Additionally, there were several genes that anti-VISTA/15Gy increased more than isotype/15Gy, such as CD247 and FasL. ** = p* < 0.05.

## Data Availability

Not applicable.

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
