# Peer review of "Enhancement of Radiation Therapy through Blockade of the Immune Checkpoint, V-domain Ig Suppressor of T Cell Activation (VISTA), in Melanoma and Adenocarcinoma Murine Models"

_ijms, 2023, doi:10.3390/ijms241813742_

Round 1

Reviewer 1 Report

This study describes the combination of radiotherapy and blockade of the immune checkpoint VISTA in two cancer murine models. Several treatments combined radiation and immune checkpoint inhibition studies have been already published, although this new study explores for the first time the comparison of two VISTA-blockading methods combined with RT. The study validates the effects on two cancer types (melanoma and adenocarcinoma) previously unexamined in the context of VISTA-enhanced RT.  Abstract is correct, results are clearly presented, and the methods and statistical analysis are appropriate. It is concluded that VISTA blockade significantly enhances the therapeutic effect of radiation in two murine tumor models.

Minor points to be corrected:

Title: Replace adiation by radiation

Figure 1: What is the meaning of the horizontal brackets inside the figure?

Line 249: Replace 2 x 106 by 2 x 106

Line 251: Replace 110 (+/-20) mm3 by 110 (+/-20) mm3

Line 274: Replace ug and uL by mg and mL (twice) .

Author Response

Reviewer 1 Feedback

This study describes the combination of radiotherapy and blockade of the immune checkpoint VISTA in two cancer murine models. Several treatments combined radiation and immune checkpoint inhibition studies have been already published, although this new study explores for the first time the comparison of two VISTA-blockading methods combined with RT. The study validates the effects on two cancer types (melanoma and adenocarcinoma) previously unexamined in the context of VISTA-enhanced RT.  Abstract is correct, results are clearly presented, and the methods and statistical analysis are appropriate. It is concluded that VISTA blockade significantly enhances the therapeutic effect of radiation in two murine tumor models. 

Minor points to be corrected:

Title: Replace adiation by radiation

Figure 1: What is the meaning of the horizontal brackets inside the figure?

Line 249: Replace 2 x 106 by 2 x 10

Line 251: Replace 110 (+/-20) mm3 by 110 (+/-20) mm3

Line 274: Replace ug and uL by mg and mL (twice) .

Response:

Thank you for your feedback. All points have been corrected in the manuscript. The horizontal brackets in Figure 1 were remnant of an old figure that surfaced during the file conversion process and have been removed. 

Reviewer 2 Report

The topic is interesting. The manuscript is quite well written. I have several suggestions:

1) Introduction: Radiation therapy (RT) has recently demonstrated promise at stimulating an enhanced immune response. The recent success of immunotherapies, such as checkpoint inhibitors, CART cells, and other immune modulators, affords new opportunities for combination with radiation. Methods: In this study, VISTA, an immune checkpoint, was blocked either by genetic knockout (KO) or an inhibitory antibody and combined with RT in two syngeneic murine cancer models. Selected mRNA, immune cell infiltration, and tumor growth delay were used to assess the biological effects. Please, improve the description of study aim and underline the novelty of the paper.

2) Results: When combined with a single 15 Gy radiation doses, both VISTA blockade techniques significantly enhanced anti-tumor immune signaling pathways at the mRNA, cellular infiltration level and tumor growth delay. The data suggests that the mechanism behind the enhanced tumor control is primarily a result of increased apoptosis and immune mediated cytotoxicity. Conclusion: VISTA blockade significantly enhances the anti-tumor effect of a single dose of 15 Gy radiation through increased expression and stimulation of cell mediated apoptosis pathways. These results suggest that VISTA is a biologically relevant immune promoter that has the potential to enhance the efficacy of a large single radiation dose in a synergic manner. Please, add some statistically significant values to support the results and the conclusions.

3) L65-66. Furthermore, this study is unique in its validations  used to assess two cancer types previously unexamined in the context of VISTA-enhanced RT. Could you please undeline the aim of the study and the novelty of the work.

4) 2. Results. Could you please underline the most important statistically significant values to support the data.

5) 3. Discussion L167-169. Radiation therapy (RT) is an effective cancer therapy. However, due to a variety of patient and normal tissue complications and its stimulation of both pro- and anti-tumor signaling, it is often not curative. Please, summarise here the most important statistically significant results of the study.

6) L231-234. In conclusion, these experiments have demonstrated VISTA blockade significantly enhances the therapeutic effect of radiation in two murine tumor models. Although many  mechanistic and confirmatory concepts remain to be determined and proven, this foundational study demonstrates the VISTA enhancement of RT is through increased immune mediated pathways, such as T-cell induced apoptosis. Please, underline the novelty of the study and the limitations of the study. Furthermore, underline the possible clinical implications of the study. 

Author Response

Reviewer 2 Feedback

The topic is interesting. The manuscript is quite well written. I have several suggestions:

Comment 1) Introduction: Radiation therapy (RT) has recently demonstrated promise at stimulating an enhanced immune response. The recent success of immunotherapies, such as checkpoint inhibitors, CART cells, and other immune modulators, affords new opportunities for combination with radiation. Methods: In this study, VISTA, an immune checkpoint, was blocked either by genetic knockout (KO) or an inhibitory antibody and combined with RT in two syngeneic murine cancer models. Selected mRNA, immune cell infiltration, and tumor growth delay were used to assess the biological effects. Please, improve the description of study aim and underline the novelty of the paper.

1a) Response: Thank you for your feedback. The study aims and novelty have been clarified in both the abstract and the introduction. Briefly, the aim of this study is the evaluate whether and to what extent VISTA blockade can potentiate the tumor control ability of radiation therapy at the same dose, and to evaluate the potential underlying mechanisms of this enhancement. As for the novelty of the study, although several promising combined radiation and immune checkpoint inhibition studies have been published, our study is novel in that it is the first comparison of two VISTA-blocking methods (antibody inhibition and genetic knockout) in combination with RT. Furthermore, our study is unique in its use of NanoString gene expression analyses to characterize the underlying mechanisms of RT enhancement by VISTA blockade.

Comment 2) Results: When combined with a single 15 Gy radiation doses, both VISTA blockade techniques significantly enhanced anti-tumor immune signaling pathways at the mRNA, cellular infiltration level and tumor growth delay. The data suggests that the mechanism behind the enhanced tumor control is primarily a result of increased apoptosis and immune mediated cytotoxicity. Conclusion: VISTA blockade significantly enhances the anti-tumor effect of a single dose of 15 Gy radiation through increased expression and stimulation of cell mediated apoptosis pathways. These results suggest that VISTA is a biologically relevant immune promoter that has the potential to enhance the efficacy of a large single radiation dose in a synergic manner. Please, add some statistically significant values to support the results and the conclusions.

2a) Response: Thank you for your feedback. We have included the relevant statististically significant values to both abstract and result sections. In summary, when combined with a single 15 Gy radiation dose, VISTA blockade via genetic knockout in the B16 model and via anti-VISTA antibodies in the MC38 model significantly improved survival compared to RT alone by an average of 5.5 and 6.3 days, respectively (P<0.05).

Comment 3) L65-66. Furthermore, this study is unique in its validations used to assess two cancer types previously unexamined in the context of VISTA-enhanced RT. Could you please undeline the aim of the study and the novelty of the work.

3a) Response: Thank you for your feedback. As noted in 1a, the study aims and novelty have been clarified in both the abstract and the introduction.

Comment 4) Could you please underline the most important statistically significant values to support the data

4a) Response: Thank you for your feedback. A paragraph has been added at the end of manuscript section 2.2 summarizing the most important statistically significant values that support the data.

Comment 5) Discussion L167-169. Radiation therapy (RT) is an effective cancer therapy. However, due to a variety of patient and normal tissue complications and its stimulation of both pro- and anti-tumor signaling, it is often not curative. Please, summarise here the most important statistically significant results of the study.

5a) Response: Thank you for your feedback. Similar to 2a and 4a, the most important statistically significant results have been summarized in the discussion section.

Comment 6) L231-234. In conclusion, these experiments have demonstrated VISTA blockade significantly enhances the therapeutic effect of radiation in two murine tumor models. Although many mechanistic and confirmatory concepts remain to be determined and proven, this foundational study demonstrates the VISTA enhancement of RT is through increased immune mediated pathways, such as T-cell induced apoptosis. Please, underline the novelty of the study and the limitations of the study. Furthermore, underline the possible clinical implications of the study. 

6a) Response: Thank you for your feedback. We have added a new paragraph and thoroughly edited the two penultimate paraphs of our conclusion to address the novelty, limitations, and clinical relevance of our findings (edited manuscript lines 233-240 and 249-256)

Round 2

Reviewer 2 Report

The manuscript has been improved as requested. No further comments